# Alteration in Tracheal Morphology and Transcriptomic Features in Calves After Infection with *Mycoplasma bovis*

**DOI:** 10.3390/microorganisms13020442

**Published:** 2025-02-18

**Authors:** Fan Liu, Fei Yang, Lei Guo, Mengmeng Yang, Yong Li, Jidong Li, Yanan Guo, Shenghu He

**Affiliations:** 1College of Animal Science and Technology, Ningxia University, Yinchuan 750021, China; liuf2785@163.com (F.L.); yangfeiweiwuxian@126.com (F.Y.); guoleinxyc@163.com (L.G.); 15809605110@163.com (M.Y.); lijidongi@foxmail.com (J.L.); 2Institute of Animal Sciences, Ningxia Academy of Agricultural and Forestry Sciences, Yinchuan 750002, China; 3College of Life Science and Technology, Ningxia Polytechnic, Yinchuan 750002, China; 13309518987@189.cn

**Keywords:** *Mycoplasma bovis*, trachea, transcriptome, signaling pathway, IL-6

## Abstract

*Mycoplasma bovis* is one of the most important pathogens in animal husbandry, and the current infection and morbidity rates are increasing year by year, causing great losses to the farming industry and seriously affecting animal welfare. In this study, we took tracheal tissues from calves infected with *M. bovis* to make pathological tissue sections for observation, and selected tracheal tissues for transcriptome sequencing to screen differentially expressed genes based on the threshold |log2FoldChange| > 1 and Padjust < 0.05 and functional enrichment, to explore in depth the potential mechanisms of bovine tracheal damage caused by bovine tracheitis. Experiments were conducted to observe the changes in tracheal tissues after *M. bovis* infection through pathological sections of the trachea of *M. bovis*-infected calves. From the transcriptome sequencing results, we mined the main differential genes and important metabolic pathways of *M. bovis* causing damage to the trachea of calves. It was found that the cricoid cartilage tissue of the trachea was congested and hemorrhagic after *M. bovis* infection in calves, and the pathological sections showed localized necrosis of epithelial cells, disorganization, high inflammatory cell infiltration in the interepithelial and lamina propria, and some epithelial cell detachment. Transcriptome sequencing identified 4199 DEGs, including 1378 up-regulated genes and 2821 down-regulated genes. KEGG enrichment analysis indicated that the differential genes were enriched to 59 significantly differing signaling pathways, and a number of important metabolic pathways related to tracheitis induced by *M. bovis*-infected calves were unearthed. The major ones included IL-17, the Toll-like receptor, JAK/STAT, the PI3K-Akt signaling pathway, etc. In this study, we found that *M. bovis* infection of calves caused inflammatory damage to the trachea, and transcriptome sequencing results also showed significant differences in the expression of key genes such as IL-6 inflammatory factor, CASP8, and APOA1.

## 1. Introduction

*Mycoplasma bovis* is an environmental pathogen that infects bovine animals and was first isolated from milk during a mastitis outbreak in the United States in 1961 [1]. It is highly contagious and can cause pneumonia and mastitis; in severe cases, it can also cause arthritis, otitis media, meningitis, myocarditis, and even reproductive disorders in cattle [2,3,4]. Prevention and control of bovine mycoplasmosis are hampered by the lack of efficient vaccines and proven treatment protocols, leading to a global spread of this pathogen, which poses a major threat to animal health and agricultural welfare [4,5].

Tracheal tissue is an important part of the respiratory system, and tracheal epithelial cells and lymphocytes form the tracheal epithelial barrier that is the first line of defense against the invasion of respiratory air pathogens [6]. Numerous studies have shown that inflammation caused by infecting pathogenic bacteria is one of the key factors leading to the destruction of tracheal tissue [7]. One of the most important modes of infection transmission of *M. bovis*, an environmental pathogen, is through airborne entry into the epithelial cells of the bovine respiratory tract to colonize and re-invade the lungs, affecting the respiratory system of cattle [8,9]. Cattle infected with *M. bovis* show symptoms such as coughing, wheezing, and bronchitis, and in severe cases, death can even occur [10]. Currently, *M. bovis* causes significant economic losses to the dairy industry and beef cattle producers each year and is also considered one of the most economically impactful infectious diseases [11,12]. The molecular mechanisms by which *M. bovis* damages host organisms are not fully understood due to its small genome, lack of metabolic pathways, and the absence of a cell wall specific for natural resistance to certain antibiotics. The trachea, as the primary organ of the respiratory system, may be the first to be affected in environmental infections with *Mycoplasma bovis*. Whereas the transition of the pathogen from infecting the host to presenting a disease state is a complex biological process, genomic changes in the host after the onset of infection are in turn a prerequisite for disease damage. Therefore, understanding the genomic changes in the host after disease onset is essential for understanding pathogen–host interactions to prevent and treat disease progression.

The transcriptome represents the full range of expressed RNA transcripts and can be the first to reflect the corresponding environmental and potential genetic and other influences on change in an organism [13]. With the rapid development of high-throughput sequencing technology in recent years, transcriptome research has been widely used in the life sciences and other fields [14], and has also been of great advantage in providing host immune responses to microbial infections [15]. Signaling pathways, as carriers of intercellular information transfer, are an effective way to understand changes in organisms and regulation of cellular life activities. Gene function and signaling pathways can more clearly reveal the link between molecular regulatory mechanisms and phenotypes. Revealing changes in different genes and signaling pathways through transcriptional information is an excellent avenue for understanding the occurrence of diseases in organisms. Healthy immune development of calves interacts with their mothers [16], and calves as a reserve are an important part of the sustainability of the cattle breeding process.

In this study, we performed replication of a pathological model of *M. bovis* infection and observed and further confirmed the inflammatory damage to tracheal tissues by *Mycoplasma bovis* from histological changes. In addition, transcriptome analysis identified differentially expressed genes (DEGs) in cattle with *M. bovis* infection degrees that are involved in multiple classical pathways associated with inflammatory injury, and explored potential modes of action. These findings deepen our understanding of the molecular pathogenesis of tracheal injury in *M. bovis*-infected calves and provide new insights for subsequent investigation of disease control.

## 2. Materials and Methods

### 2.1. Establishment of the Animal Model

*M. bovis* isolate NX114 was resuscitated by PPLO liquid broth medium (BD DifcoTM, San Jose, CA, USA) and then passaged and cultured to enrich the bacterial solution up to the concentration of 1 × 10^10^ CFU/mL, which were used as inoculums. Six healthy Holstein calves of the same weight and size at 60 days of age, negative for *M. bovis* pathogen by nasal swab, were selected and randomized into a blank control group (group C, *n* = 6), and a pathology model group (group P, *n* = 6). After 1 week of acclimatization feeding, calves in group P were subjected to replication of the pathology model through the infection route of *M. bovis* bacterial solution nose drops at a specific dose of 2 mL/day for 3 consecutive days, and in group C, an equal amount of PBS solution (Biosharp, Hefei, China) was operated in the same manner and at the same time for 3 consecutive days.

Animal experimental protocols and operations were conducted with reference to the requirements of the Guidelines for Ethical Review of Laboratory Animal Welfare in the People’s Republic of China [17], and the experimental animals were approved by the Animal Research Ethics Committee of Ningxia University (No. NXU-2022-097).

### 2.2. Sample Collection

All animals were kept in a controlled environment until the end of euthanasia by bloodletting of calves after anesthesia via sodium pentobarbital. The tracheal tissues of calves in group C and group P were collected aseptically on the 7th day after the end of infection, and the tissues were rapidly frozen in liquid nitrogen after being put into the freezing tubes, and then transferred to a −80 °C freezer for subsequent experiments after the tissues were completely frozen.

### 2.3. Histopathological Examination

Appropriate amounts of tracheal tissues from group C and group P were taken and pathology sections were made by a pathology examination system of orientations and practices (SOP) procedure. The specific procedures were dehydration with ethanol solution, tissue trimming, paraffin embedding, sectioning, hematoxylin and eosin (H&E) staining, and sealing of sections. The pathological tissue changes were observed under the microscope, and then the images of the tissue sections were captured using a digital section scanner (Pannoramic 250, 3DHISTECH Ltd., Budapest, Hungary).

### 2.4. Transcriptome Sequencing and Quality Control

An appropriate amount of frozen calf tracheal tissue was liquid-nitrogen-frozen and then ground into powder form to extract total RNA from the tissue samples by the Trizol Reagent (Invitrogen Life Technologies, Thermo Scientific, Bremen, Germany). The concentration and the pure of the extracted RNA were examined by using an ultramicro spectrophotometer (Nanodrop 2000, Thermo Fisher Scientific, Waltham, MA, USA) and agarose gel electrophoresis to detect the integrity of the RNA, and the fragment analyzer (Bioanalyzer 2100 system, Agilent, Palo Alto, CA, USA) to determine the value of the RNA. Sequencing libraries were constructed using only high-quality RNA samples (total RNA ≥ 1 μg, OD260/280 = 1.8~2.2, concentration ≥ 35 ng/μL). The mRNA was isolated from total RNA by A-T base pairing with ployA using magnetic beads with Oligo (dT). Using Illumina second-generation sequencing technology, mRNA was enriched and then processed on a sequencer platform (Illumina Novaseq 6000, Shanghai Personal Biotechnology Cp., Ltd., Shanghai, China) to isolate small fragments of about 300 bp by magnetic bead screening. The cDNA was synthetic by transcribing in reverse and sequenced on the platform. The results obtained from sequencing were compared with the database, and the clean reads were compared with the reference genome through post-quality control data, while the mapped reads were obtained for subsequent analyses, as well as an assessment of the quality of the results of this sequencing comparison. Sequence comparison analysis was performed using Hisat2 (http://ccb.jhu.edu/software/hisat2/index.shtml, accessed on 2 February 2023) software [18].

Reference gene source: Bos_taurus;

Reference genome version: ARS-UCD1.2; Reference genome version: http://asia.ensembl.org/Bos_taurus/Info/Index, accessed on 2 February 2023;

Utilizing expression quantification software RSEM (http://deweylab.biostat.wisc.edu/rsem/, accessed on 2 February 2023), the quantitative index was TPM, and the gene and transcript expression levels were quantified as the number of transcript entries counted, respectively. The transcriptome data were screened by analyzing raw counts using negative binomial distribution-based difference analysis software DESeq2 (http://bioconductor.org/packages/stats/bioc/DESeq2/, accessed on 2 February 2023) with Padjust < 0.05 & |log2FC| ≥ 1 as the criterion, while the expression differential analysis was performed through a BH multiple test correction method for data processing.

### 2.5. RT-qPCR Validation of the DEGs Results

A total of 12 DEGs with significant fold change were chosen at random and subjected to RT-qPCR to validate the RNA-seq results. Total RNA was extracted from two sets of tracheal tissue samples by using Trizol reagent (Invitrogen Life Technologies), referring to the manufacturer’s instructions. Quantification of total RNA was conducted by spectrophotometer (Merinton SMA4000, Meilin Hengtong (Beijing) Instrument Co., Ltd., Beijing, China). The mRNA was reverse transcribed to cDNA by PrimeScript^TM^ RTreagentKit with gDNA Eraser (TaKaRa, Code No. RR047A, Beijing, China) at 37 °C for 15 min, 85 °C for 5 s, and stored at 4 °C. TB Green^®^ Premix Ex Taq^TM^ II FAST qPCR (TaKaRa, Code No. CN830A, Beijing, China) was selected for quantitative PCR amplification, and the sequences of the primers synthesized by the Shanghai Bioengineering Co., Ltd. (Shanghai, China) are shown in Table 1. Glyceraldehyde 3-phosphate dehydrogenase (GAPDH) was used as a reference gene to normalize the data [19]. The relative expression levels of different genes were used for calculation by the 2^−ΔΔCT^ method.

## 3. Results

### 3.1. Abnormal Tracheal Tissue in Calves Due to M. bovis Infection

Observations on different groups of tracheal tissue initially after calf dissection. There were significant differences in the tracheal tissue of calves in the two groups following *M. bovis* infection. Group C had normal tracheal tissue and moist mucosa; Group P had marked congestion in the cricoid cartilage space of the trachea and marked congestion in the mucosal layer (Figure 1A,B).

To verify the effect of *M. bovis* on tracheal tissues, hematoxylin and eosin (H&E) staining of tracheal tissues from infected calves was performed. Tissue sections of infected calves showed a clear structure of the tracheal mucosa, submucosa and outer mucosa in group C as compared to the control group; the mucosal epithelium was a pseudo-ciliated columnar epithelium, with intact cilia and a variable number of cup-shaped epithelial cells; the submucosal layer was a loose connective tissue, with no obvious demarcation from the lamina propria, and a small number of glands locally, while the mesenchyme had an infiltration of inflammatory cells, with a predominance of lymphocytes; no obvious pathologic changes were seen (Figure 2A,B). In group P, the tracheal tissue showed more epithelial cell necrosis and disordered arrangement, as well as shedding of epithelial ciliary structures while the nuclei of necrotic epithelial cells were solidified, disintegrated, cytoplasmic lysis, and the epithelial cells were detached locally; the inter-epithelial cells and the lamina propria were infiltrated with more inflammatory cells, and neutrophils were predominantly present (Figure 2C,D).

### 3.2. Sequencing Data Quality Results

To further investigate the key molecular and molecular mechanisms of inflammatory injury in the trachea of calves infected with *M. bovis*, we chose to analyze a total of six samples of calf trachea from group C and group P for transcriptomic analysis. A total of 48.70 Gb of clean data were obtained, and the clean data of each sample were more than 6.14 Gb. The proportion of Q30 bases was above 93.39%, and the comparison with the reference genome was above 95.05%. In addition, according to the principles of random sequence interruption and double-strand complementarity, the base pair GC and AT contents of each sequencing read should be similar and stable throughout the sequencing process, and the GC contents of the six samples in this result were stable between 51.1 and 53.01%. The GC content of the six samples was stable at 51%, which indicates that the sequencing results are of good quality and can be used for the next step of analysis. The detailed results are presented in Table 2, and the credibility of the sequencing results is in line with the next step of bioinformatic analysis.

### 3.3. Transcriptome Sequencing Analysis

Before comparison of differentially expressed genes (DEGs) between groups P and C, principal component analysis (PCA) was used to evaluate the diversity and similarity of replicates within and between groups (PC1: 71.05% and PC2: 16.21%), which showed that biological replicates in the same group were tightly clustered together, and samples from different treatment groups were clearly separated, which indicates that the sequencing data of samples in the groups had a high degree of similarity and significant biological reproducibility of the sequencing data of the samples within the group, and the correlation between the groups was low with significant differences, confirming the reliability and stability of the sequencing results (Figure 3A). The levels of gene expression of the six samples of group C and group P were similar to the box plot demonstration (Figure 3B). Based on the thresholds (|log2FoldChange| > 1 and Padjust < 0.05), comparison of group P with group C yielded 4199 DEGs, 1378 up-regulated DEGs and 2821 down-regulated DEGs. The histogram of expression differences (Figure 3C) and scatter plot (Figure 3D) show the distribution of differential genes in the two groups.

### 3.4. qRT-PCR Verification

Corroboration of the reliability of transcriptome sequencing results. A total of 12 differentially expressed genes, including APOA1, BLA-DQB, CXCL8, MMP3, and NFκB1, were selected randomly for expression verification by qRT-PCR, and the results showed that the expression of the 12 differentially expressed genes was up- and down-regulated in a trend consistent with the results of transcriptome sequencing. Although the CXCL8 validation results showed non-significant differences, the trend of the expression changes was uniform with the trend of the sequencing results (Figure 4).

### 3.5. DEGs Functional Annotations

In order to better understand the potential regulatory mechanisms underlying the inflammatory damage caused by *M. bovis* infection of calf trachea, we performed a comparative group P and group C transcriptomic analysis to explore the changes in the expression of relevant genes at the transcriptional level in the host after infection with *M. bovis*, a pathogenic microorganism. GO annotation and KEGG annotation were first performed on 4199 DEGs to gain a preliminary understanding of the functions these DEGs play in the life process. DEGs were annotated to 55 GO entries, of which 1378 up-regulated DEGs were enriched to 52 GO entries, which were assigned to 22 biological process (BP) entries, 15 cellular components (CCs), and 15 molecular functions (MFs), respectively; and 2821 down-regulated DEGs were enriched to 53 GO entries, which comprised 24 biological process (BP) entries, 14 cellular components (CCs), and 15 molecular functions (MFs) (Appendix A). The top three terms of up-regulated and down-regulated genes annotated were the BP category, in which the top three terms were cellular process, biological regulation, and metabolic process (Figure 5A). The KEGG functional annotation results of DEGs showed a total of 344 pathways enriched in six major KEGG metabolic pathway branches, including a cellular process, environmental information processing, genetic information processing, and an organismal biological system, as well as human disease and metabolism. The results of KEGG functional annotation of DEGs showed that a total of 344 pathways were enriched in six KEGG metabolic pathway branches, including the cellular process, environmental information processing, genetic information processing, organismal biosystems, and human disease and metabolism, among which up-regulated DEGs were annotated to 322 pathways, and down-regulated DEGs were annotated to 340 pathways, while the top three most frequently annotated DEG terms were signal transduction, immune system, infectious disease: viral. Bar graph showing the top 30 significantly enriched KEGG pathways (Figure 5B).

### 3.6. Functional Enrichment of DEGs

A comprehensive analysis of GO enrichment of DEGs from *M. bovis*-infected calf tracheal tissues was performed with the aim of elucidating the functions and biological pathways of the major exercises associated with the altered genes. The results of GO functional enrichment showed that a total of 1188 GO entries were enriched in the DEGs, with BP accounting for the largest proportion of the entries (77.6%), CC accounting for 12.3%, and MF accounting for 10.1% (Appendix A). All differential gene enrichment of the first 30 GO entries was BP, mainly including inner dynein arm assembly (GO:0036159), epithelial cilium movement involved in extracellular fluid movement (GO:0003351), cellular response to mechanical stimulus (GO:0071260), and neutrophil migration (GO:1990266) (Figure 6A). The GO enrichment results (Padjest ≤ 0.05) indicate that many bioprocess changes occurred at the transcriptome level in the tracheal tissues of *M. bovis*-infected calves, and many of the entries coincided with the defense response after pathogenic bacterial infection, such as neutrophil chemotaxis and migration, suggesting a systemic immune-inflammatory response of the organism, which is in line with the previous results of the pathological tissue sections that also show an inflammatory injury of the tissues in detail, and part of the epithelial cell detachment accompanied by the infiltration of neutrophils. 

The KEGG function was enriched (Padjest ≤ 0.05) with 59 significantly different KEGG pathway IDs distributed across six species categories, including 25 Human Diseases (42%), 12 Organismal Systems (20%), 8 Environmental Information Processings (14%), 3 Cellular Processes (5.0%), 10 Metabolisms (17%) and 1 Genetic Information Processing (2%) (Appendix A). The most important enrichment pathways include the IL-17 signaling pathway, the cytokine-cytokine receptor interaction, the PPAR signaling pathway, and so on. The top 30 KEGG-enriched pathways are shown in the bar graph (Figure 7A). At the same time, we constructed enrichment chordal maps to depict the top 20 enriched pathways with their corresponding differential genes, and the analysis shows that DEGs such as IL6 and APOA1 were enriched (Figure 7B).

To further explicitly probe changes in damage caused by *M. bovis* infection of calf trachea at the transcriptional level, we again performed a topological analysis of the set of differential genes belonging to Organismal Systems and Environmental Information Processing, which showed that these genes are associated with the immune system, signaling and signaling molecules interacting with the KEGG pathway. The gene set enriched the top 20 KEGG pathways related to immune injury such as the PI3K-Akt signaling pathway, the JAK-STAT signaling pathway, the Toll-like receptor signaling pathway and many other signaling pathways related to immune injury; we constructed a network analysis map of multiple pathway interactions (Figure 8A). The dynamic mulberry diagram demonstrates the enrichment pathway corresponding to some differential genes and expression (Figure 8B).

The experimental results also show that many immune cytokines as well as key genes for disease damage were differentially expressed, and several related classical pathways were also enriched. DEGs are involved in the regulation of a wide range of physiological and pathological processes, and different DEGs in turn directly or indirectly regulate different signaling pathways, which trigger interactions with multiple signaling pathways (Figure 9).

## 4. Discussion

*M. bovis* is an important group of bovine pathogens affecting animal productivity and healthy welfare farming [20], and it is very common for *M. bovis* infections to trigger lung inflammation and ultimately lung tissue damage [21]. Laboratory tests for bronchopneumonia with intercurrent pneumonia in beef cattle show the highest detection rate of *M. bovis*, which exhibits tissue immune cell infiltration and bronchial epithelial cell detachment [22]. Most descriptions of the pathologic changes in the fine bronchioles of calf pneumonia caused by *M. bovis* are that the fine bronchioles contain neutrophils and some monocytes in the lumen, and that the whole is surrounded by a large number of inflammatory cells [23,24]. Respiratory tract injury in cattle by *M. bovis* has been reported frequently, but reports of studies targeting tracheal inflammation have not been found.

### 4.1. M. bovis Infection Causes an Inflammatory Tissue Injury Response

The trachea, an important component of the respiratory system, is also an immunologically active barrier surface that plays a protective role in the organism’s response to infection by pathogenic bacteria [25]. The main manifestation of airway inflammation is epithelial cell damage and an increase in the number of inflammatory cells [26]. In this experiment, dissection and comparison of calf tracheal tissues before and after infection revealed that calf tracheal tissues showed obvious pathological changes after *M. bovis* infection, with severe congestion in the cricoid cartilage space. Histopathological damage to tracheal epithelial cells was also obvious in group P compared with group C. Some epithelial cell necrosis, detachment, and inflammatory cell infiltration was clearly seen in pathological sections, indicating that *M. bovis* infection caused tracheal inflammation and tracheal injury destroyed the normal function of the tracheal mucosal barrier.

### 4.2. M. bovis Infection Causes Crosstalk Expression of Multiple Pathways Associated with Inflammation

Our experimental results demonstrate that *M. bovis* infection of the calf tracheal tissue affects calves at the transcriptional level, with an enrichment of many signaling pathways and interconnected crosstalk of multiple pathways associated with inflammation. *M. bovis* infection is characterized by an inflammatory response and involves multiple strategies to counteract the host immune response, so the host naturally corresponds to a number of immunomodulatory changes [27]. The IL-17 signaling pathway is a key regulatory pathway in respiratory diseases and its expression is upregulated by airway inflammation and tracheal epithelial cell injury [28]. Our study found that the IL-17 signaling pathway was also the most prominent signaling pathway belonging to the immune system enriched by DEGs after *M. bovis* infection, and the pathway enrichment included 39 differential genes such as IL-6, IL1β, and CASP8. In a recent study, a group herbal medicine was found to inhibit the development of *M. bovis* pneumonia probably by modulating the PI3K-Akt and IL-17 signaling pathways, which were also enriched in our trial of *M. bovis* calf tracheal inflammation [29]. The importance of the IL-17 signaling pathway in linking to tracheal inflammation caused by *M. bovis*-infected calves is again illustrated. The PI3K-Akt signaling pathway is associated with infectious diseases and innate immunity, supports cell metabolism, survival and reproduction, and is a key pathway for feedback of environmental information and regulation of the organism’s response to injury [30,31]. The PI3K-Akt signaling pathway is also an upstream participant in the activation of STAT3 [32], which has also been shown in previous studies to affect macrophages and promote airway inflammation [33]. Previous researchers analyzed molecular changes in *M. bovis*-positive breast tissues and concluded that the PI3K-Akt signaling pathway and the NF-kB signaling pathway were associated with *M. bovis* infection in positive breast tissues [34]. An established in vitro model of *M. bovis* infecting BMECs revealed that autophagy was inhibited through a PI3K-Akt pathway-dependent pathway after *M. bovis* infection with BMECs [35]. This pathway was similarly significantly enriched in this assay, suggesting that activation of the PI3K-Akt signaling pathway by *M. bovis* infection may also cause the suppression of autophagy in tracheal tissue cells, thereby enhancing the inflammatory response.

The Toll-like receptor signaling pathway, as a specific family pattern recognition receptor responsible for microbial pathogens and generating an innate immune response, was also a close fit between the significance of the differences in this enrichment and the content of the experiments [36]. Previous studies have also shown that *M. bovis* infection can activate the Toll-like receptor signaling pathway in BMECs [37]. Our results show that calf tracheal tissues revealed significantly elevated expression levels of TLR2, TLR4, TLR7, and TLR10, as well as MyD88 after infection with *M. bovis*, and initiated the innate immunity Toll-like receptor signaling pathway. At the same time, activation of TLRs initiates downstream signaling such as the JAK/STAT cascade, which regulates intracellular kinases to further stimulate inflammatory and antigen-specific immune responses [38]. JAK and STAT are also known as key components of many signaling pathways that regulate cell growth, differentiation, survival and pathogen resistance and, as a family of IL-6 receptors, synergistically regulate B-cell differentiation, plasma cell genesis and the acute-phase response [39]. A positive correlation with the activation of the JAK/STAT signaling pathway in sepsis and allergic contact dermatitis has been reported in TLR4 expression [40]. This pathway is associated with many immune function-related genes, and inflammatory genes such as IL-1β, IL-6, and IL-8 were significantly upregulated in the same infections [41]. The present study is consistent with many previous findings that *M. bovis* infection causing tracheal injury is accompanied by a significant enrichment of JAK/STAT pathway differences and a significant up-regulation of several differential genes related to immunity and inflammation such as IL-1β and IL-6.

### 4.3. Widespread Effects of M. bovis Infection on Gene Expression

Numerous studies have shown that IL-6 is able to function as an anti-inflammatory early marker that promotes the secretion of other inflammatory factors and amplifies the inflammatory response [42], and its expression level increases after the infection of pathogenic microorganisms [43]. Other research showed that mycoplasma is the main cause of airway inflammation and that neutrophils as well as pro-inflammatory cytokines such as IL-6, IL-8, and IL-1β are also significantly increased [44]. The present transcriptional findings are consistent with many of the previous results of Gondaira: multiple inflammatory factors such as IL-1β, IL-6, and IL-8 were upregulated and increased in expression after *M. bovis* infection [45,46,47].

CASP8 was previously identified as an exogenous cell death initiation factor that plays a role not only in apoptosis and necrosis, but also in the maintenance of tissue homeostasis, and acts as a scaffolding protein to promote cytokine production [48]. More recent studies have even identified this gene as central to pan-apoptosis, and it has been experimentally demonstrated that CASP8 plays a key role in maintaining endothelial integrity by preventing both inflammation and tissue hemorrhage [49]. One noteworthy point is that our statistical enrichment pathway and differential gene correspondence mulberry diagrams show a breakdown in CASP8 occupancy, and the specific results of transcriptomics show that CASP8 is not only significantly up-regulated in the differential genes, but also enriched in 12 signaling pathways with significant differentials. It is very likely that CASP8 is one of the key genes responsible for the tracheal inflammation caused by *M. bovis*.

There are also key genes such as APOA1 that are negatively correlated with disease risk levels [50]. Some cases have also found that pathogens infecting the liver may produce and release large amounts of cytokines such as IL-6, and that these highly elevated levels of cytokines may instead lead to liver injury, which in turn affects the synthetic methylpentanediol acyl-coenzyme A, ultimately resulting in a reduction in APOA1 synthesis [51]. APOA1 changes accordingly with the inflammatory response and also correlates with disease severity [52]. APOA1 was similarly enriched after *M. bovis* infection as a result of the present study, and there was a significant difference in expression as a result of RT-qPCR validation. Group C had low IL-6 expression and high APOA1 expression when not infected with the pathogen, and the two were inversely adjusted after infection in line with the results of the previous investigations.

Overall, we targeted differences in gene expression and host-driven biological systems from the *M. bovis* infection of calf trachea to explore potential mechanisms of damage from *M. bovis* infection. These changes may help us to gain insight into the pathways of injury of this pathogen. Although our experiments identified over 4000 differential genes, they reinforce the fact that *M. bovis* infection of calves is a complex physiological process. Significant differences in gene expression mediate effects on multiple biological processes related to immune inflammation, among others, and cannot be singularly clustered in a single differential gene or pathway. To our knowledge, there are currently no studies that have uncovered disease damage by comparing transcriptomic changes in different tracheal tissues. It is easy to imagine that the process of pathogen infection is accompanied by a myriad of genes that are differentially expressed, signaling, and a severely dysregulated immune response ultimately leading to host inflammatory disease. Moreover, the inflammatory response is usually a key factor in the pathologic progression of organ diseases [53].

## 5. Conclusions

In the present research, we used histologic observations to determine that *M. bovis* causes calf tracheal inflammatory injury. More than 4000 DEGs were identified using transcriptomics data to determine the severe effects of *M. bovis* spongiformis infection on calf tracheal tissues at the molecular level and enriched several important metabolic pathways associated with pathogen-infected calf tracheitis, mainly including the IL-17 signaling pathway, the Toll-like receptor signaling pathway, and the PI3K-Akt signaling pathway, which are associated with inflammatory injury. Notably, IL-6 inflammatory factor, CASP8, and APOA1 may also be the key genes with altered expression after pathogen infection, which are closely related to the inflammatory response to tracheal injury in calves. Based on the research results of the previous and current experiments, we found that inflammatory damage is one of the key mechanisms for the pathogenesis of *M. bovis* infection. Therefore, reducing inflammatory damage in animals may become a research direction for treating *M. bovis* infections, which provides theoretical and technical support for comprehensively exploring the characteristics of *M. bovis* and understanding the diseases caused by the infection of this pathogen.

## Figures and Tables

**Figure 1 microorganisms-13-00442-f001:**
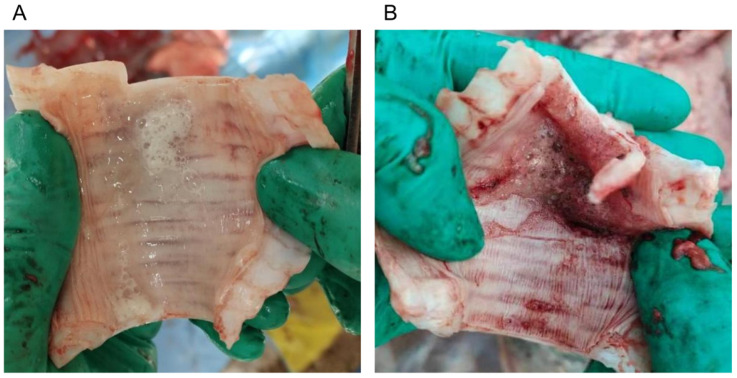
(**A**) Anatomical diagram of normal tracheal tissue; (**B**) Tracheal tissue showed obvious hemorrhage and congestion.

**Figure 2 microorganisms-13-00442-f002:**
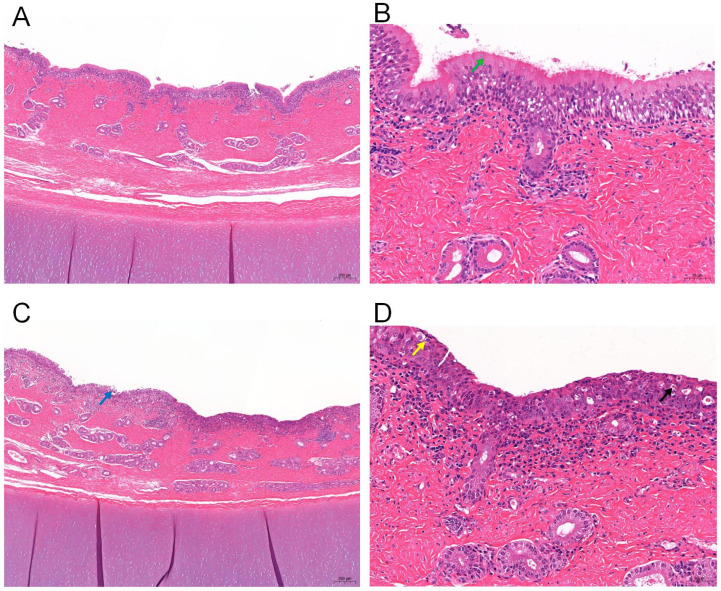
Micrographs of H&E staining of calf trachea (**A**,**C**) 40×; (**B**,**D**) 200×. Effect of *M. bovis* infection on histopathology of trachea in calf. (**A**,**B**) Normal calves’ trachea tissue, Ciliated structure of normal pseudo-ciliated columnar epithelium (green arrow); (**C**,**D**) Histopathological changes in trachea in calf. shedding of epithelial cells (blue arrows); Neutrophilic granulocyte infiltration (yellow arrows); Epithelial cells necrosis (black arrows).

**Figure 3 microorganisms-13-00442-f003:**
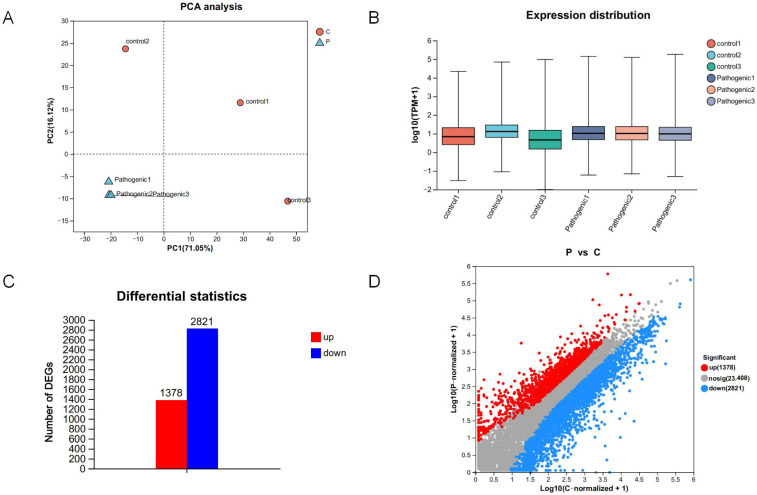
(**A**) PCA score of transcriptome analysis of tracheal tissue of calves. The horizontal axis represents the contribution degree of principal component 1 (PC1) to the distinguished samples, and the vertical axis represents the contribution degree of principal component 2 (PC2) to the distinguished samples. (**B**) The box diagram shows the distribution of gene expression among different samples. The ordinate is the value of expression after log10 logarithmic processing, and the horizontal line in the diagram indicates the median gene expression in the sample. (**C**) Histogram of DEGs in the pathogenic group and control group. (**D**) Scatter plot of differentially expressed genes: with red dots (up) and blue dots (down) indicating significant gene differences (*p* < 0.05), and gray dots indicating non-differential genes. Using a fold change cutoff of 2.0.

**Figure 4 microorganisms-13-00442-f004:**
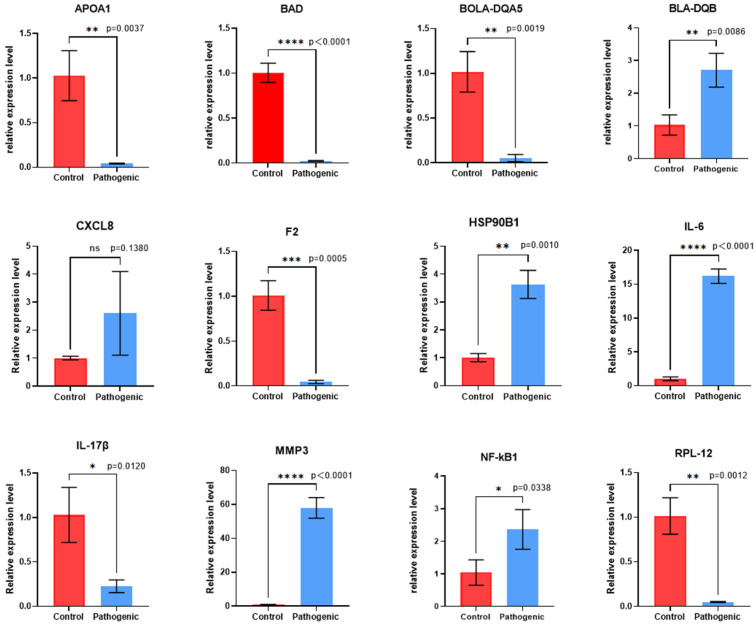
Relative expression of differential genes. Significance was tested by *t*-test, with *p* < 0.05 indicated by *, *p* < 0.01 indicated by **, *p* < 0.001 indicated by ***, *p* < 0.0001 indicated by ****, and *p* > 0.05 (not significant) indicated by NS. Shows the average extent of differential genes in the two groups, with horizontal coordinates indicating different groups and vertical coordinates indicating relative expression.

**Figure 5 microorganisms-13-00442-f005:**
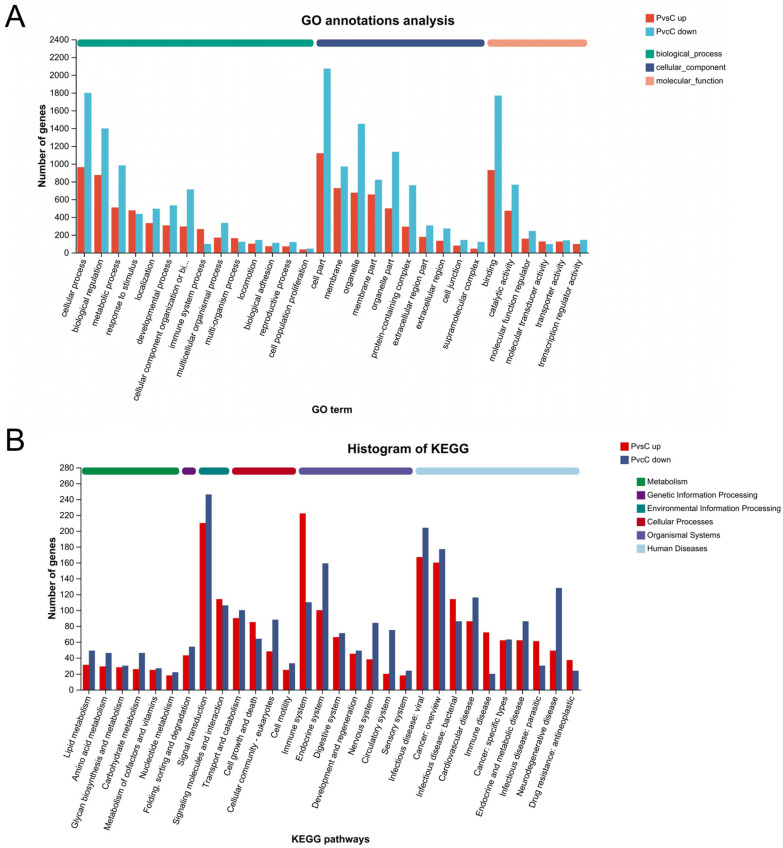
(**A**) GO annotations analysis of DEGs. Shows the top 30 enriched GO terms of DEGs. Red represents up-regulated genes, blue represents down-regulated genes. (**B**) KEGG annotations analysis of DEGs. Shows the top 30 enriched KEGG pathway of DEGs. The abscissa is the name of the KEGG metabolic pathway; The ordinate is the number of genes annotated to this pathway.

**Figure 6 microorganisms-13-00442-f006:**
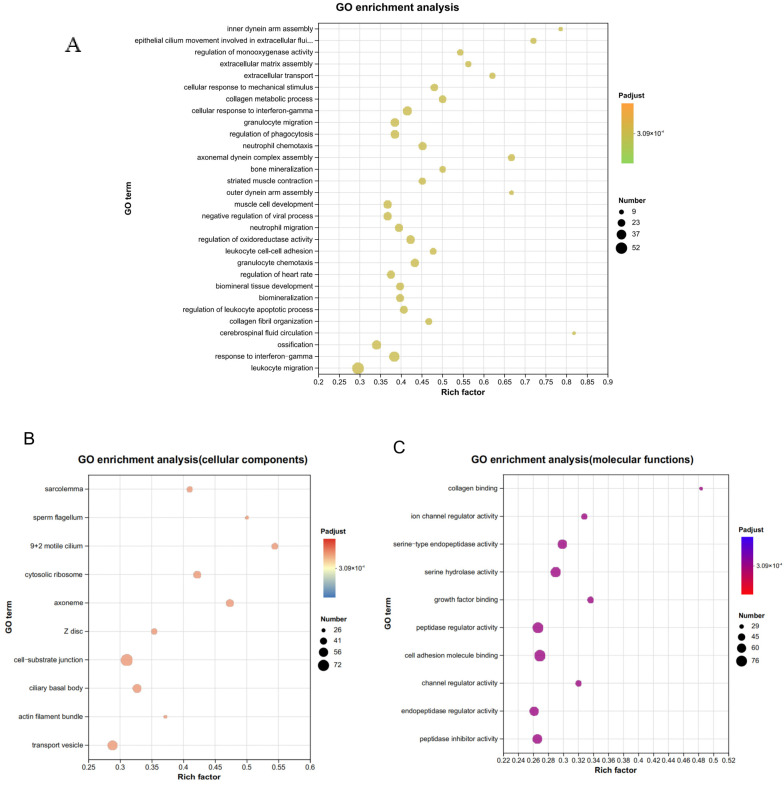
(**A**) GO enrichment analysis of DEGs. (**A**) The top 30 GO enrichment analysis of DEGs. (**B**) GO enrichment analysis in the top 10 of the CC category. (**C**) GO enrichment analysis in the top 10 of the MF category.

**Figure 7 microorganisms-13-00442-f007:**
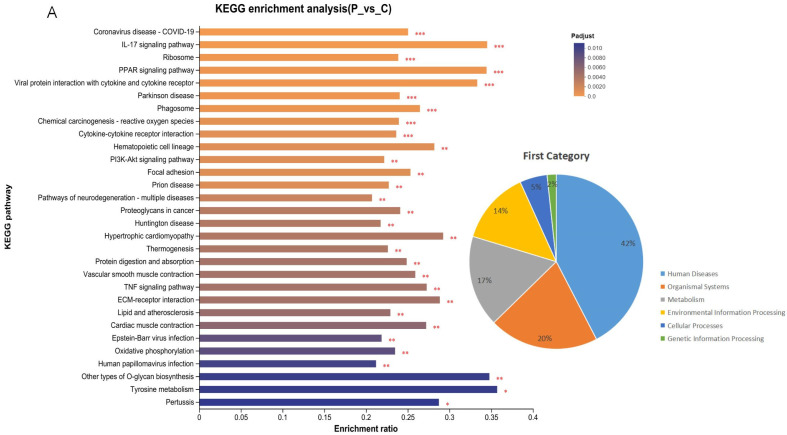
KEGG pathway enrichment analysis of DEGs. (**A**) The top 30 enriched pathways of DEGs. Column color gradients indicate the significance of enrichment, with *p* < 0.001 indicated by ***, *p* < 0.01 indicated by **, *p* < 0.05 indicated by *. (**B**) KEGG pathway enrichment chord diagram: Differentially expressed genes correspond to significantly enriched pathways. Shows the correspondence between DEGs and the KEGG pathway.

**Figure 8 microorganisms-13-00442-f008:**
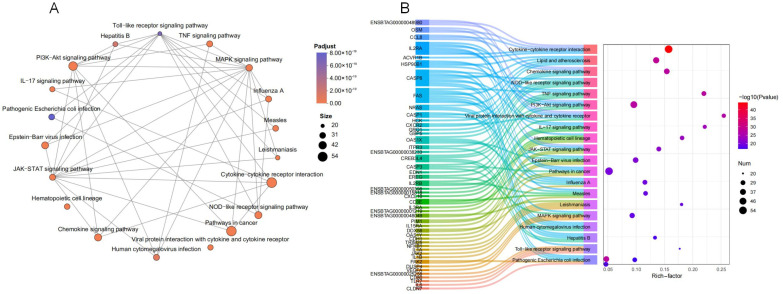
(**A**) Enrichment network analysis of 20 KEGG pathways. (**B**) Dynamic mulberry diagram of the DEGs and KEGG enrichment pathway.

**Figure 9 microorganisms-13-00442-f009:**
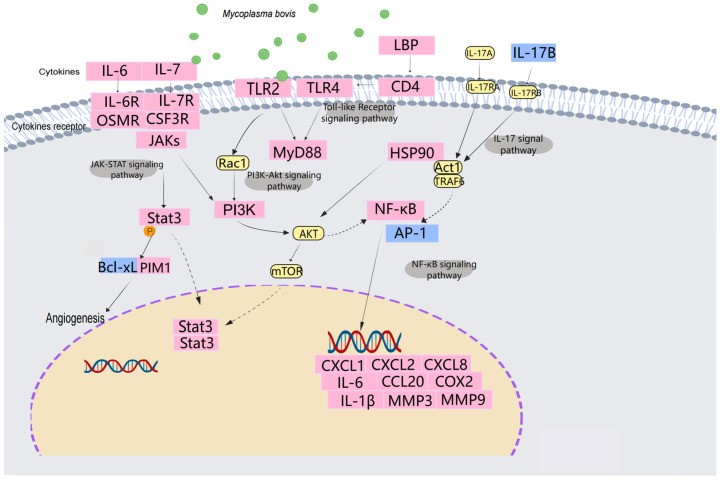
Selected differentially significant signaling pathways in *M. bovis*-infected bovine trachea. Pink nodes represent significantly differentially up-regulated genes, yellow nodes represent non-significantly differentially up-regulated genes, and blue represents down-regulated genes.

**Table 1 microorganisms-13-00442-t001:** List of primers used for quantitative RT-PCR validation.

Name	Forward Primer (5′-3′)	Reverse Primer (5′-3′)	Gene ID
APOA1	GCTGACCTTGGCTGTGCTCTTC	TTCACCCGATCCCAGGATGACTG	Gene ID:281631
BAD	TTGAGCAGAGTGAGCAGGAAGAC	TTAGCCAGTGCTTGCTGAGACC	Gene ID:615013
BLA-DQB	CTGGACAGCAGTTGTGATGGTG	CGAAATCCTTTGGCGAGTCTCTG	Gene ID:539241
BOLA-DQA5	TGTTTTCCAAGTCTCCCGTGATGC	TGTGACCGAGTGCCCGTTCC	Gene ID:282494
CXCL8	AGCTGGCTGTTGCTCTCTTGG	TGGGGTGGAAAGGTGTGGAATG	Gene ID:280828
F2	AGGTACAACTGGAAGGAGAATCTGG	CTTGGCTGCTGTCTGCTTGTC	Gene ID:280685
HSP90B1	AAGATCGAGAAGGCTGTGGTGTC	ATGTCCTTGCCTGTCTGGTATGC	Gene ID:282646
IL-6	TGATGAGTGTGAAAGCAGCAAGG	GCAGTGGTTCTAATCAAGCAAATC	Gene ID:280826
IL-17B	TCCTTCTCACCATCTCCATCTTCC	ACACCAGGTCCAGTGGCAAC	Gene ID:504992
MMP3	ACGGCATTCAGTTCCTGTACGG	GGGTTCGGGAGGCACAGATTC	Gene ID:281309
NF-κB1	TGACTACGCAGTGACAGGAGAC	TATGAAGGTGGATGATTGCCAAGTG	Gene ID: 616115
RPL12	TGATGACATCGCCAAGGCAACTG	TGGGCTTGTCTGTTCTGAATGGTC	Gene ID: 404133
GAPDH	GTCTTCACTACCATGGAGAAGG	TACTGGATGACCTTGGCCAG	Gene ID:281181

**Table 2 microorganisms-13-00442-t002:** List of transcriptome data quality assessments.

Sample	Raw Reads	Raw Bases	Clean Reads	Clean Bases	Q30(%)	GC Content(%)	Total Reads	Total Mapped
Control1	43,580,784	6,580,698,384	42,696,550	6,208,009,189	93.41	51.90	42,696,550	41,203,330 (96.5%)
Control2	67,488,566	10,190,773,466	66,820,766	9,701,663,897	94.20	52.08	66,820,766	64,238,447 (96.14%)
Control3	43,950,424	6,636,514,024	42,724,002	6,144,103,535	93.39	53.01	42,724,002	40,717,074 (95.3%)
Pathogenic1	52,807,056	7,973,865,456	52,214,386	7,706,211,884	93.92	51.69	52,214,386	50,067,909 (95.89%)
Pathogenic2	71,991,514	10,870,718,614	71,327,596	10,469,575,414	94.27	51.93	71,327,596	67,795,966 (95.05%)
Pathogenic3	58,290,528	8,801,869,728	57,533,278	8,466,256,638	94.23	51.14	57,533,278	55,061,299 (95.7%)

## Data Availability

Data will be made available on request.

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
