# Peer review of "Alteration in Tracheal Morphology and Transcriptomic Features in Calves After Infection with Mycoplasma bovis"

_microorganisms, 2025, doi:10.3390/microorganisms13020442_

Round 1

Reviewer 1 Report

Comments and Suggestions for Authors

Dear authors,

the manuscript „Alteration of tracheal morphology and transcriptomic features in calves after infection with Mycoplasma bovis” is generally properly written. The manuscript will be adequate for publishing in the journal if authors follow the minor comments:

Authors should check their manuscripts for spaces between words and be careful with periods in sentences, as there are several of them unnecessary in the middle of sentence.

Write name of bacteria and “in vitro” with italic.

Line 19: write “Pathological” with small letter

Lines 25-26: Change the sentence – write “signaling pathway” once.

Line 36: Add references. In the ref. 2 there is information about otitis only.

Line 127: Delete [16] because it is twice.

Line 139: Expand the abbreviation "DEGs".

Lines 350-353:This fragment should be delated.

Line 439: Correct the fragment “production In”.

Author Response

Dear Reviewer:

Thank you very much for taking the time to review this manuscript.

 In the following, we detail our point-by-point responses to your comments and suggestions.

Comments 1: Authors should check their manuscripts for spaces between words and be careful with periods in sentences, as there are several of them unnecessary in the middle of sentence.

Response 1: Thank you for pointing this out. We agree with this comment. We checked the whole text and adjusted to remove some unnecessary extra periods in some of the sentences.

Comments 2: Write name of bacteria and “in vitro” with italic.

Response 2: Thank you for pointing this out. We agree with this comment. There are a few instances where “Mycoplasma bovis” is not italicized in the manuscripts , and we have checked for this throughout the article and have italicized all bacterial names, The “in vitro” in line 385 of the manuscript has also been italicized.

Comments 3: Line 19: write “Pathological” with small letter

Response 3: Thanks for your helpful comments. We agree with this comment. In line 19, “Pathological” has been changed to “pathological”.Due to a partial revision of the manuscript, this content is currently on line 21.

Comments 4: Lines 25-26: Change the sentence – write “signaling pathway” once.

Response 4: Thank you for pointing this out. We agree with this comment. We have deleted the repetition of “signaling pathway” in lines 26-27 and retained “signaling pathway” only once.

Comments 5: Line 36: Add references. In the ref. 2 there is information about otitis only.

Response 5: Thank you for pointing this out. We agree with this comment.On line 36 we added 2 references to describe the information that Mycoplasma bovis infection can cause arthritis, meningitis, myocarditis and even reproductive disorders

Comments 6: Line 127: Delete [16] because it is twice.

Response 6: Thank you for pointing this out. We agree with this comment.We fixed the situation where reference [16] appeared twice on line 127 by adding 2 references on line 36 of the article, so that line 127 now shows [17]

Comments 7: Line 139: Expand the abbreviation "DEGs".

Response 7: Thank you for pointing this out. We agree with this comment.Differentially expressed genes appears for the first time on line 80, but we forgot to add the abbreviation “DEGs”, so we add (DEGs) on line 80. Hopefully our changes will be recognized.

Comments 8: Lines 350-353:This fragment should be delated.

Response 8: Thank you for pointing this out. We agree with this comment.Lines 350-353 were mistakes we made in formatting the article, and they have been deleted in the new manuscript.

Comments 9: Line 439: Correct the fragment “production In”.

Response 9: Thank you for pointing this out. We agree with this comment. The sentence in line 439 has been adjusted to“CASP8 was previously identified as an exogenous cell death initiation factor that plays a role not only in apoptosis and necrosis, but also in the maintenance of tissue homeostasis and acts as a scaffolding protein to promote cytokine production”.The content is in lines 424-426.

Reviewer 2 Report

Comments and Suggestions for Authors

Alteration of tracheal morphology and transcriptomic features in calves after infection with Mycoplasma bovis

Title: I guess the format should be capital wordings

Abstract:

More details on the methodology and statistical analysis are needed.

L26-27: Instead of discussion-like sentences, please provide a conclusion in one sentence.

Introduction:

L37: Please avoid repetition: "...widespread spread..." does not flow well

The hypothesis(s) are not clearly stated.

M&M:

Subheadings should be capitalized.

L127: Ref {16} stated twice

Results:

Subheadings should be capitalized

If possible, please provide p value separatly for each figure graph (where applicable) under the figure or above, or provide superscripts on columns in the related figure (if significant)

Discussion:

The outcome of the study was discussed well.

Conclusion:

One sentence at the end to suggest a future direction would enrich the section.

Author Response

Dear Reviewer:

Thank you very much for taking the time to review this manuscript.

 In the following, we detail our point-by-point responses to your comments and suggestions.

Comments 1: Title: I guess the format should be capital wordings

Response 1:Thank you for pointing this out. We agree with this comment. We've changed the title format to capital wordings.

Comments 2: Abstract: More details on the methodology and statistical analysis are needed.

Response 2: Thank you for pointing this out. We agree with this comment. We have added a description of the methods and statistics to the manuscript abstract, specifically in lines 12-16, which hopefully completes the description of our experiments.

Comments 3: Abstract:  L26-27: Instead of discussion-like sentences, please provide a conclusion in one sentence.

Response 3: Thank you for pointing this out. We agree with this comment. The sentence in lines 26-27, where we have adjusted the expression, has been modified to a single sentence conclusion. "In this study, we found that M. bovis infection of calves caused inflammatory damage to the trachea, and transcriptome sequencing results also showed significant differences in the expres-sion of key genes such as IL-6 inflammatory factor, CASP8, and APOA1.."

Comments 4: Introduction:  L37: Please avoid repetition: "...widespread spread..." does not flow well

Response 4: Thank you for pointing this out. We agree with this comment. We've changed the "...widespread spread..."  to "... global spread..." in line 37.This sentence is in lines 40-41 of the revised manuscript, and your suggestion to avoid a poorly worded sentence is much appreciated.

Comments 5: Introduction:  The hypothesis(s) are not clearly stated.

Response 5: Thank you for pointing this out. We agree with this comment. We hypothesize in lines 56-62 of the introductory section that the trachea may be the earliest tracheal depiction to be affected by Mycoplasma bovis infections, which hopefully adds to the increased completeness of the manuscript.

Comments 6: M&M:  Subheadings should be capitalized.

Response 6: Thank you for pointing this out. We agree with this comment. We changed all the subtitles of the manuscript to capitalization.

Comments 7: M&M:  L127: Ref {16} stated twice

Response 7: Thank you for pointing this out. We agree with this comment.Our mistake caused {16} to appear twice, and the redundant {16} has been deleted in the new manuscript, and since a new reference was added earlier, the {16} in the 134 lines was turned into {17}. 

Comments 8: Results: Subheadings should be capitalized

Response 8: Thank you for pointing this out. We agree with this comment. We have capitalized all subheadings of results in the manuscript.

Comments 9: If possible, please provide p value separatly for each figure graph (where applicable) under the figure or above, or provide superscripts on columns in the related figure (if significant)

Response 9: Thank you for pointing this out. We agree with this comment. We added the p-values for each data in the upper column of the important Figure 4.It is true that it is not convenient to represent all the p-values in the other plots in the figure, but we have also provided p-values for GO enrichment and KEGG enrichment in the accompanying table.

Comments 10: Conclusion: One sentence at the end to suggest a future direction would enrich the section.

Response 10: Thank you for pointing this out. We agree with this comment.We have adjusted the expression in the last sentence of the article's conclusion based on your valuable comments. At the same time, we suggest that the reduction of “inflammatory damage” can be one of the research directions in the later stage, and the specific adjustments have been corrected in red in the manuscript.

Reviewer 3 Report

Comments and Suggestions for Authors

The article is original and very relevant for the field. The authors have investigated the mechanisms related to the damage of bovine trachea caused by M. bovis through histopathologic observation and transcriptome sequencing. Experiments were conducted to observe the changes of tracheal tissues after M. bovis infection through histological sections of the trachea of M. bovis-infected calves, followed by transcriptome sequencing.

The methology of the study is very modern, from histopathology to most modern molecular biology, bioinformatics and biostatistics tools.

The results showed that the cricoid cartilage tissue of the trachea was congested and hemorrhagic after M. bovis infection in calves, and the histopathological sections showed localized necrosis of epithelial cells, disorganization, high inflammatory cell infiltration in the interepithelial and lamina propria, and some epithelial cell detachment. Transcriptome sequencing identified 4,199 differentially expressed genes, including 1,378 up-regulated genes and 2,821 down-regulated genes. Authors found that IL-6 inflammatory factor, CASP8 and APOA1 could be the key genes with altered expression after pathogen infection, which are closely related to the inflammatory response to tracheal injury in calves. This results could be very important for public health, in the context of increasing antibiotic-resistance of different bacterial strains, providing a theoretical basis for future understanding of the mechanism of Mycoplasma spp respiratory injury and treatment of inflammation caused by this pathogen.

The conclusions are consistent with the evidence and arguments presented.

The references are appropriate, including some very relevant authors experience in the field.

I recommend some minor corrections.

1.     For histopathological studies PAS stain and a better magnification would be verry usefull

For histopathological methodology or Discussions authors may see also

Experimental Insights on the Use of Secukinumab and Magnolol in Acute Respiratory Diseases in Mice. WOS:001276485100001, https://doi.org/10.3390/biomedicines12071538,

Detailed minor corrections

-line 99- poison attack? Correct is infection

Line 139 DEG- give detailed name at first use

Line 176, Fig 1A- is not histopathological, but macroscopical

In Fig 2D legend authors should mention also the lose of cilli from the surface of epithelium. Consultancy of a Veterinary pathologist should be very usefull

Fig 6 B,C and 7, B,C should be bigger

Lines 350-353- Delete the paragraph

Author Response

Dear Reviewer:

Thank you very much for taking the time to review this manuscript.

 In the following, we detail our point-by-point responses to your comments and suggestions.

Comments 1: For histopathological studies PAS stain and a better magnification would be verry usefull/ For histopathological methodology or Discussions authors may see also   Experimental Insights on the Use of Secukinumab and Magnolol in Acute Respiratory Diseases in Mice. WOS:001276485100001, https://doi.org/10.3390/biomedicines12071538,

Response 1: Thank you for pointing this out. We agree with this comment. We appreciate your valuable comments. We chose only one HE stain for histopathological observation because this staining method is the easiest for us to follow with all the materials available. After obtaining the staining results and finding that they were consistent with the expected results and pathological changes were observed, the next step of the experiment was carried out. We also read the article you recommended and very much agree that for histopathological studies, PAS staining and better magnification will be very useful, and will add the use of PAS staining for histopathological observations in our subsequent studies, thank you very much for your guidance and suggestions.

Comments 2: line 99- poison attack? Correct is infection

Response 2: Thank you for pointing this out. We agree with this comment.We have corrected the inappropriate expression “poison attack” in line 99 to “infection”.

Comments 3: Line 139 DEG- give detailed name at first use

Response 3: Thank you for pointing this out. We agree with this comment.Differentially expressed genes first appeared in line 80 and we forgot to add the abbreviation “DEGs” for which we apologize, so we added (DEGs) in line 80. No changes were made in line 139.

Comments 4: Line 176, Fig 1A- is not histopathological, but macroscopical

Response 4: Thank you for pointing this out. We agree with this comment.The misrepresentation of  “Histopathological changes of trachea in calves” were error in our labeling, which we have removed.

Comments 5: In Fig 2D legend authors should mention also the lose of cilli from the surface of epithelium. Consultancy of a Veterinary pathologist should be very usefull

Response 5: Thank you for pointing this out. We agree with this comment.Thank you for your valuable and professional opinion. In Fig. 2B we added the labeling of normal ciliated structures, and for clarity, the description that normal epithelial cells have ciliated structures is also presented again in line 172 of the article.

Comments 6: Fig 6 B,C and 7, B,C should be bigger

Response 6:Thank you for pointing this out. We agree with this comment.We've resized Figures 6 and 7 to make them all bigger, hopefully to your satisfaction.

Comments 7: Lines 350-353- Delete the paragraph

Response 7: Thank you for pointing this out. We agree with this comment.Lines 350-353 were mistakes we made in formatting the article, and they have been deleted in the new manuscript.

Round 2

Reviewer 2 Report

Comments and Suggestions for Authors

Dear Authors,

Thank you for taking the time and effort to respond to all my comments in detail.

I can see significant improvements in the revision file and now can accpet this article in the presented form.

Good luck